# Multi-Modality Cell Segmentation based on nnU-Net Pipeline

**Haotian Lu**
School of Electronic Information
and Electrical Engineering
Shanghai Jiao Tong University
Shanghai, Dongchuan Rd. 800
flick-lu@sjtu.edu.cn

**Jinghao Feng**
School of Electronic Information
and Electrical Engineering
Shanghai Jiao Tong University
Shanghai, Dongchuan Rd. 800
fjh1345528968@sjtu.edu.cn

**Zelin Peng**
School of Electronic Information
and Electrical Engineering
Shanghai Jiao Tong University
Shanghai, Dongchuan Rd. 800
godlin_bd@126.com

**Wei Shen**[*]
School of Electronic Information
and Electrical Engineering
Shanghai Jiao Tong University
Shanghai, Dongchuan Rd. 800
wei.shen@sjtu.edu.cn

## Abstract

Cell segmentation is an important upstream task in Medical Image Analysis. Recent years, data-driven deep learning methods made ground-breaking achievements. In this challenge, multi-modal and partially-labeled dataset is provided. On the dataset, this paper proposes a multi-modality cell segmentation framework based on nnU-Net pipeline and iterative self-training method. Our model reaches 0.6101 mean F1 score on tuning set.

## 1   Introduction

Because of sensitive nature and privacy concerns of Medical Image Analysis, researchers usually get dispersive, sometimes multi-source samples. Besides, it's extremely time-consuming for doctors to accurately annotate the images. These lead to a multi-modality and semi-supervised segmentation task. As the samples are independently collected from different centers, we need to address big modality gap. We also want our model be robust to new modalities. These requires high-level comprehension from our model. Besides, less than half of the cases are labeled, so semi-supervised methods must be applied to utilize the unlabeled samples. We first construct a fully-supervised model on labeled samples using nnU-Net [1]. To further improve boundary performance, we apply weight map [2]. After fine-tuning, we fuse the model into an iterative self-traning framework [3] to make use of the unlabeled cases. Finally, we convert our semantic segmentation results to instance segmentation result depending on the connectivity between the pixels.

The remainder of this paper includes: **2**.Method that introduces our method in detail, **3**.Experiments that shows experimental details, **4**.Results and discussion that shows the model performance and further discussions and **5**.Conclusions.

36th Conference on Neural Information Processing Systems (NeurIPS 2022).

## 2 Method

We apply nnU-Net [1] as the backbone network of our approach. We first fine-tune u-net on the labeled samples. To improve boundary performance, weight map [2] is used to emphasize boundary loss. We further utilize the unlabeled samples with self-training-based semi-supervision framework [3].

### 2.1 Preprocessing

Following preprocessing methods provided officially, we first create interior maps with instance segmentation masks. Specifically, we assign cell interior with 1, background with 0 and boundary with 2. By this means, we formulate our task as a pixel-wise 3-class classification problem. By differentiating boundary and interior pixels, we hope to detect overlapping cells in instance segmentation phase.

Second, we perform channel normalization on sample images to alleviate modality gaps. Given a 3-channel image, for each channel, if not empty, we linearly re-scale pixel intensities into interval $0 \sim 255$. Finally, to adapt to nnU-Net, we convert sample images and masks to .nifti format.

### 2.2 Proposed Method

In recent years, convolutional neural networks significantly promotes medical image segmentation. In 2015, U-Net [2] was proposed, and its variants kept improving segmentation performance in medical image settings. In 2018, nnU-Net [1], which integrates certain network designing principles, achieved state-of-the-art performance without manual tuning. Due to its self-adapting property and multiple data augmentations, we build our model upon nnU-Net. To optimize boundary performance, weight map [2] is applied to encourage loss's attention on boundary quality. As a whole, the above network is further utilized in self-training-based semi-supervision framework, our strategy to use unlabeled cases.

### 2.3 Network Architecture

Our model bases on a generic U-Net, with skip connections between different resolution stages of encoder and decoder. Both encoder and decoder are stacks of convolutional blocks. In the encoder, 9 convolutional blocks are used. Each block can further decompose into two sets of consecutive $3 \times 3$ convolution layers followed by instance normalization and leaky ReLU activation. Dropouts are included. In the decoder, 8 convolutional blocks are used, each takes the output of the previous block, upsamples feature maps with $2 \times 2$ transposed convolution layer, and concatenates the output with corresponding shortcut output from the encoder. Afterwards, the concatenated features are fed into a convolutional block similar to the blocks in the encoder.

### 2.4 Weight Map

Due to modality variety, it's challenging to detect cell boundary accurately. To address this, we adapt weight map method proposed in [2] to emphasize boundary loss, thus forcing the network learn more about boundary features. A weight map of a case is an array with the same size as the case image, which gives a weight for the loss of each pixel. To emphasize boundary areas, we give boundary pixels bigger weights than background and cell interior. Following [2], we define the weight map of a case $\mathbf{x}$ as

$$w(\mathbf{x}) = w_c(\mathbf{x}) + w_0 \cdot exp\left( - \frac{(d_1(\mathbf{x}) + d_2(\mathbf{x}))^2}{2\sigma^2} \right)$$

where $w_c(\mathbf{x})$ denotes the class-balanced map, where we assign each pixel with the inverse of the area that the class of the pixel accounts for. $d_1(\mathbf{x})$ and $d_2(\mathbf{x})$ denotes the distance to the boundary of the nearest and the second nearest cell respectively. We set $w_0 = 10$ and $\sigma = 5$. In training phase, we compute weight maps beforehand to prevent extra time cost.

### 2.5 Loss function

Our loss is composed of the unweighted summation between Dice loss and cross-entropy loss, as it's proved robust to use compound loss functions in various medical image segmentation tasks [4].

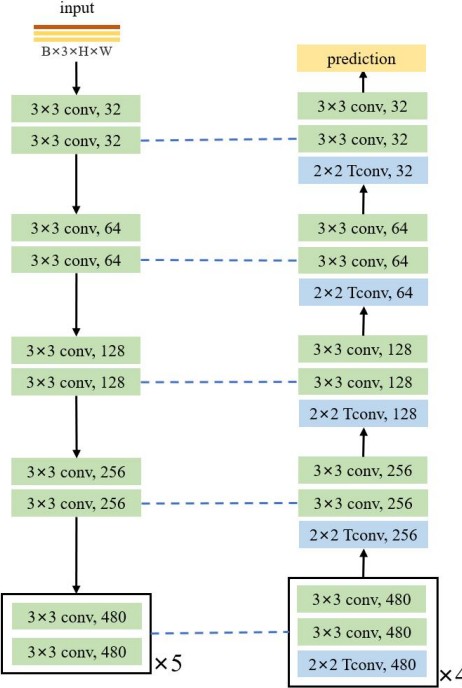

Figure 1: Network architecture. in every convolution block, the kernel size and output channel number is given. 'Tconv' denotes transposed convolution layer. The arrows denote data flows, and dotted lines shows skip connections between encoder and decoder.

Following [1], we use the multi-class version of the Dice loss variant proposed in [5], which is defined as

$$L_{Dice}(u, y) = -\frac{2}{|C|} \sum_c^C \frac{\sum_i^N u_{i,c} y_{i,c}}{\sum_i^N u_{i,c} + \sum_i^N y_{i,c}}$$

The cross-entropy loss can be written as

$$L_{CE}(y, \hat{y}) = -\frac{1}{N} \sum_i^N \sum_c^C \mathbb{I}_{y_{i,c}=1} log(\hat{y}_{i,c})$$

where $N$ denotes the total pixel number of a case, $C$ denotes number of classes. $y_{i,c}$ is the one-hot encoding of the ground truth mask. $\hat{y}_{i,c}$ denotes the network output, the logit probability of that pixel i belongs to class c. And $u_{i,c}$ is the softmax output of $\hat{y}_{i,c}$.

The Dice loss is weighted in response to weight map method. When calculating true positive, true negative, false positive and false negative arrays, we multiply them with the weight map in pixel-wise manner.

The total loss function is

$$L_{Total} = L_{Dice} + L_{CE}$$

## 2.6 Semi-supervision strategy

To utilize the unlabeled cases, we apply iterative self-training framework [3]. We fine-tune U-Net with weight map on the labeled cases. The learned network, commonly refered to as 'teacher network', then generates pseudo-annotations for the unlabeled cases. The pseudo-labels are thresholded into one-hot vectors. With labeled and pseudo-labeled cases, we re-train the teacher network, deriving 'student network'. The re-training process is then repeated iteratively.

## 2.7 Result conversion

In this section, we convert our semantic segmentation results to instance segmentation result. Specifically, we divide the cell part into different connected blocks and label them according to connectivity between the pixels. In the same time, we will convert the nnU-Net result format to standard result format.

# 3 Experiments

## 3.1 Dataset

We only use the official training set to train our model. The labeled images are used to train basic nnU-Net model and the unlabeled images are used to train final nnU-Net model.

## 3.2 Implementation details

### 3.2.1 Environment settings

The development environments and requirements are presented in Table 1.

Table 1: Development environments and requirements.

| System | Ubuntu 18.04.6 LTS |
|---|---|
| CPU | Intel(R) Xeon(R) Silver 4210R CPU @ 2.40GHz |
| RAM | 125GB |
| GPU (number and type) | Four NVIDIA GeForce RTX 3090 24G |
| CUDA version | 11.4 |
| Programming language | Python 3.7.13 |
| Deep learning framework | Pytorch (Torch 1.12.1, torchvision 0.13.1) |
| Code | nnU-Net |

### 3.2.2 Training protocols

Our training process consists of two parts: Baseline and Retraining. We use the Baseline model to generate template mask for unlabeled images and use them to train the Retraining model.The main training protocols are presented in Table 3 and Table 4. Our model depends on the nnU-Net baseline, so the Data augmentation and other processing method depends on it.

**Data augmentation** (Based on the winning solutions in FLARE 2021, we recommend using extensive data augmentation)

patch sampling strategy during training (e.g., randomly sample $1024 \times 1024$ patches) and inference (slide window with a patch size $1024 \times 1024$)

optimal model selection criteria

**Preprocess** We use the official method to normalize the images and convert the label to three classes: cell, boundary, background. Then we convert the image and label to format as ".nii.gz" with three modality.

**Data augmentation** All augmentation methods we used depend on nnU-Net.We randomly cut the images to fit the size $1024 \times 1024$. Then we use the following transformation methods to enhance the image including **mirror transform, gamma enhancement, rotate, scale, gauss** and so on.

**Deep Supervision** In order to train the model effectively, we use Deep Supervision to calculate the loss. For the upsampling results of each size in nnU-Net, we convert them to the output space and calculate loss between corresponding size targets and the output.

**Model Selection** All model parameters depend pn the 5-fold cross validation. When we find the best model, the nnU-Net will combine 5-fold together.Meanwhile, it will use some method like cutting small connected components yo enhance the model.

Table 2: Training protocols. If the method includes more than one model, please present this table for each model seperately.

| | |
|---|---|
| Network initialization | "he" normal initialization |
| Batch size | 2 |
| Patch size | 80×192×160 |
| Total epochs | 1000 |
| Optimizer | SGD with nesterov momentum ($\mu = 0.99$) |
| Initial learning rate (lr) | 0.01 |
| Lr decay schedule | halved by 200 epochs |
| Training time | 72.5 hours |
| Loss function | |
| Number of model parameters | 41.22M[1] |
| Number of flops | 59.32G[2] |

**Predict** After finding the best model, we can find corresponding target for testing images. We convert the result to three channel and differentiate different cells by boundary.

Table 3: Training protocols for Baseline.

| | |
|---|---|
| Network name | Baseline |
| Batch size | 2 |
| Patch size | 3×1024×1024 |
| Total epochs | 5×500 |
| Cross-Validation | 5 fold |
| Optimizer | SGD with nesterov momentum ($\mu = 0.99$) |
| Initial learning rate (lr) | 0.01 |
| Lr decay schedule | halved by 250 epochs |
| Training time | 42 hours |
| Loss function | CrossentropyLoss and DiceLoss |
| Number of model parameters | 52.56M |
| Number of flops | 262.61G |

Table 4: Training protocols Retraining.

| | |
|---|---|
| Network name | Retraining |
| Batch size | 2 |
| Patch size | 3×1024×1024 |
| Total epochs | 5×200 |
| Cross-Validation | 5 fold |
| Optimizer | SGD with nesterov momentum ($\mu = 0.99$) |
| Initial learning rate (lr) | 0.003 |
| Lr decay schedule | halved by 100 epochs |
| Training time | 16 hours |
| Loss function | CrossentropyLoss and DiceLoss |
| Number of model parameters | 52.56M |
| Number of flops | 262.61G |

# 4 Results and discussion

## 4.1 Quantitative results on tuning set

Our F1 score on tuning set is 0.6101. If we only use the labeled cases (fully-supervised), the F1 score on tuning set is 0.6021. With our semi-supervised framework, though without many iterations, the unlabeled cases do provide useful information.

## 4.2 Qualitative results on validation set

We show an example of good segmentation results and an example of bad segmentation results are shown in Fig 2 and 3.

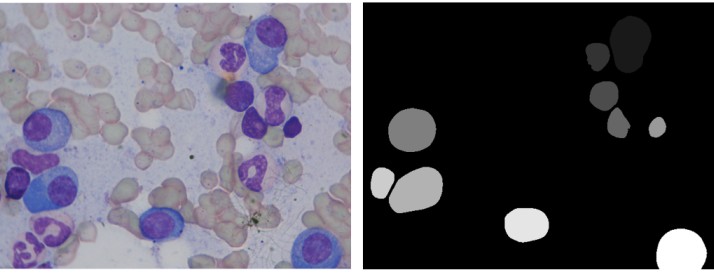

Figure 2: An example of good segmentation results.

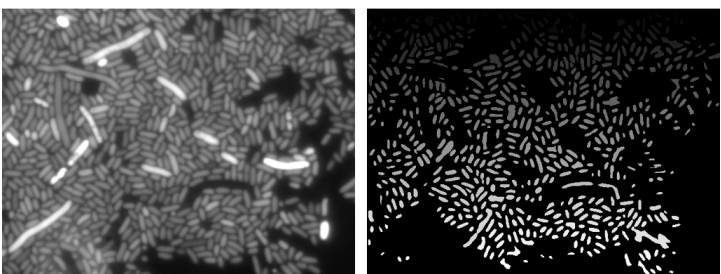

Figure 3: An example of bad segmentation results.

## 4.3 Segmentation efficiency results on validation set

The total running time on tuning set on our own work station is 656s. This time is the running time of the whole pipeline, including the time for data preprocessing and enhancementThe time of the core reasoning part is far less than this value

## 4.4 Testing Results

Testing Results are shown in table 5

Table 5: Testing Results

| Median F1-ALL | Median F1-BF | Median F1-DIC | Median F1-Fluo | Median F1-PC |
|---|---|---|---|---|
| 0.4492 | 0.5889 | 0.4183 | 0.0274 | 0.6831 |
| Mean F1-ALL | Mean F1-BF | Mean F1-DIC | Mean F1-Fluo | Mean F1-PC |
| 0.4397 | 0.6247 | 0.4203 | 0.1159 | 0.5882 |

### 4.5 Limitation and future work

The way we utilize the unlabeled cases is still naive. Self-training method is an entropy minimization problem, but lacks prior knowledge. We believe that the segmentation performance can be further improved with introduction of carefully-designed restrictions. In the future, we should look deeper into the mathematical structures of these restrictions and form new models. Besides, medical images can have very large size, which requires faster interference. We will put our efforts in the acceleration of the model in the future.

## 5 Conclusion

This paper proposes a multi-modality cell segmentation framework based on nnU-Net pipeline. We applies weight map method to improve boundary performance, and utilize the unlabeled cases with self-training method. Our model reaches 0.6101 F1 score on tuning set and

## Acknowledgement

The authors of this paper declare that the segmentation method they implemented for participation in the NeurIPS 2022 Cell Segmentation challenge has not used any private datasets other than those provided by the organizers and the official external datasets and pretrained models. The proposed solution is fully automatic without any manual intervention.

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
