# OpenReview forum: "Multi-Modality Cell Segmentation based on nnU-Net Pipeline"
_NeurIPS.cc/2022/Challenge/CellSeg — Submitted to NeurIPS CellSeg 2022_

### Official Review · Reviewer_Meyw · 2022-12-24
**Cell segmentation using nn-UNet and pseudo labeling**

**Rating:** 5
**Confidence:** 5

**Review:**

The authors first train a nn UNet on all the labeled samples in the challenges, formulating a 3 class semantic problem (background vs. cell interior vs. cell boundaries). They then extend the training to the unlabeled samples by using predictions from the first network as pseudo-labels for training a second network on both labeled and unlabeled samples. The second models (trained on actual labels and pseudo-labels) shows a small improvement in performance.

Overall the method proposed is sound, however the presentation is not clear in some cases. Before accepting the paper I would suggest to address these points:
- Misleading description of the architecture: The authors write: "In the decoder, 8 convolutional blocks are used, each [...], upsamples feature maps with 2 × 2 transposed convolution layer, [...]". This implies that the image representation is downsampled and upsampled by a factor of 2 8 times in the network, corresponding to a total scale factor of 256. This would be a much larger factor than commonly used in U-Nets for the cell segmentation and it also does not fit to the architecture shown in Figure 1, where it looks like up and downsampling happens 4 times.
- Total loss function is missing. A combination of Dice loss and Cross Entropy Loss with pixel weighting is used. A single equation that clarifies the total loss function is missing. Currently it's not quite clear how the weighting is applied.
- The description of the semi supervision strategy is missing details. In particular: are the pseudo-labels thresholded to obtain discrete one-hot encodings, or are soft pseudo-labels used?
- The training protocol description is incomplete: the paragraph "Data augmentation" is given twice with inconsistent information (in the first paragraph the authors specify a patch size of 224x224, in the second a patch size of 1024x1024.)
- Table 2 contains the C02-eq. column without a value, please provide a value or remove the column,
- The information about prediction is mssing details: the authors write: "We convert the result to three channel and differentiate different cells by boundary." Which algorithm is used for this? Connected components?

---

### Official Review · Reviewer_TmKy · 2023-01-15
**Cell segmentation in semi-supervision strategy: nnu-net and pseudo-annotations**

**Rating:** 4
**Confidence:** 4

**Review:**

The authors tried to apply nnu-net and pseudo-annotations to build a semi-supervision strategy to solve the task. The overall method is good but without obvious novelty.

1. nnu-net or nnU-Net?

The authors cited the paper from Fabian Isensee. Actually, nnU-Net has two versions. In the abstract, experiments, and conclusion sections, the authors used nnU-Net. The rest sections were in nnu-net. To avoid confusion, the authors should use the network name in the same one (i.e., nnu-net or nnU-Net).


2. some content is incomplete or not clear

  In Section 3.2.2, the content in the data augmentation part seems incomplete or duplicated. The first one mentioned the patch size 224x224, but the second one became 1024x1024.

  In Section 5, the content in the conclusion part seems incomplete.

  In Table 2, the loss function is missing. The CO2eq is missing.


3. the paper writing was of low quality

  In Section 2.2, we build our model *upon* nnu-net.

  In Section 3.2.2, our training process consists of two parts: Baseline and *Retraining*.

  In Section 4, the authors presented the results and discussions in 4 sessions. However, the content was not sufficient and couldn't provide convincing information. Especially, in Section 4.3, the description was just one short sentence.

---

### Official Review · Program_Chairs · 2023-01-16
**Multi-Modality Cell Segmentation based on nnU-Net Pipeline**

**Rating:** 6
**Confidence:** 2

**Review:**

Authors used nnU-net coupled with weightmap to generate instance segmentations.
Authors also used an iterative semi-supervised learning framework to improve the model performance by 0.008 F1.
Overall, the improvement over the provided u-net baseline was clear, with a reported F1 score of 0.6101 on the tuning set. It also successfully utilized unlabelled data.
Typos in table 2, conclusion incomplete among other minor mistakes that did not take away from the integrity of the paper
Would like to see ablation experiments

---

### Decision · Program_Chairs · 2023-01-19

Accept